# Hash-Chain-Based Cross-Regional Safety Authentication for Space-Air-Ground Integrated VANETs †

**Gege Luo [1], Mingxian Shi [2], Caidan Zhao [2],\* and Zhiyuan Shi [1]**

[1] Department of Electronic Science and Engineering, Xiamen University, Xiamen 361001, China; luogege@stu.xmu.edu.cn (G.L.); zyshi@xmu.edu.cn (Z.S.)

[2] Department of Informatics, Xiamen University, Xiamen 361001, China; 233320171153172@stu.xmu.edu.cn

\* Correspondence: zcd@xmu.edu.cn; Tel.: +86-592-258-0078

† This paper is an extended version of our paper published in IEEE International Conference on Communications.

**Abstract:** With the increasing demand for intelligent traffic management and road network intelligent information services, the vehicular ad hoc networks (VANETs) combined with information of air, space and ground have outstanding advantages in coverage, reliable transmission, and resource richness. Due to the characteristics of heterogeneous, numerous nodes, and frequent cross-network flow, the space–air–ground integrated network (SAGIN) puts forward higher requirements for security. This paper proposes a cross-regional node identity management architecture based on the hash chain, combined with radio frequency (RF) fingerprint theory, to guarantee node identity security with a non-duplicated physical information identity authentication mechanism. At the same time, the blockchain consensus mechanism is simplified to achieve block recording and verification. OMNet ++, SUMO, and Veins co-simulation platforms are used to generate transactions for cross-regional traffic flow. Based on the Hyperledger–Fabric architecture, Kafka and PBFT consensus algorithms are simulated. The simulation results show that the average delay of a single transaction generated block is about 0.9 ms, which achieves efficient and low-latency authentication.

**Keywords:** hash chain; radio frequency fingerprint; space–air–ground integrated network; vehicular ad hoc networks; identity authentication

## 1. Introduction

As an indispensable part of the intelligent transportation system, VANETs have received extensive attention from academia and industry. VANETs are distributed, self-organizing wireless networks constructed by heterogeneous vehicle-mounted entities [1] (such as vehicles and roadside units (RSUs)), which are considered to be an essential strategy for improving transportation efficiency. However, in terms of real-time interaction of complex situation information, acquisition of large-scale traffic information, and frequent massive connection, the traditional ground information network is still difficult to meet the needs of highly intelligent assisted driving [2]. Therefore, it has become an inevitable trend to integrate space-air nodes into VANETs to provide more comprehensive situation information.

The SAGIN consists of platforms of different dimensions [3], including space-based platforms composed of orbiting satellites, which mainly provides a macro and global information network for the application of VANETs and realizes the mobile internet access to large-scale vehicles. The air-based network includes a high-altitude stratospheric platform composed of airships and a low-altitude drone platform composed of quadcopters. They provide high bandwidth, low latency, and highly reliable

line-of-sight transmission for the ground information platform composed of vehicle nodes. The SAGIN structural system is complex and exists in an open communication environment, which is extremely vulnerable to eavesdropping and interception from malicious users [4]. Moreover, the large-scale cross-regional transmission in SAGIN makes the network more susceptible to threats such as data tampering at the forwarding nodes, longer transmission delays, and exhausted communication resources [5]. Given the above characteristics, in the process of aided driving through space-air nodes, the safety requirements of VANETs in an integrated environment of space–air–ground need to be considered more.

Based on the basic idea of using digital pseudonyms as unique identifiers for identity verification without using any personally identifiable information [6], most of the current key management schemes of VANETs are designed based on the public key infrastructure (PKI) [7] It requires a long authentication time, massive certificate storage, and management overhead. In [8], a dynamic key distribution scheme for VANTEs is proposed. By sending revocation messages only to vehicles with the probability of communication with the revoked vehicle, the overhead cost is reduced, and the network use rate is improved. A secure and reliable key management method protocol (SA-KMP) is proposed in [9], which distributes the repository to each vehicle and RSUs. By incorporating the authentication mechanism into the key distribution scheme, the risk of denial of service (DoS) attacks, and the complexity caused by the PKI scheme are reduced.

The high-level encryption technology also has the phenomenon of being compromised [10]. Once the above-centralized management system based on trusted third parties is attacked in the scenario of a node moving across regions, it is easy to cause communication blockage or even paralysis. Furthermore, the current PKI technologies for managing vehicle keys by different organizations in the VANETs are different and incompatible. It will be challenging for vehicles to interoperate when performing cross-regional authentication. Therefore, this paper proposes a cross-regional node identity management architecture based on the hash chain, combined with the physical identity information authentication mechanism to ensure the security of node identity. At the same time, the blockchain consensus mechanism was simplified to implement block recording and authentication, and finally, the effectiveness of the scheme was verified through simulation experiments. The contributions of this paper are as follows:

1.  Using the uniqueness and non-repudiation of RF fingerprints, the physical layer features are used as the identity of the vehicle node, so that it cannot be tampered with or forged.
2.  We Simplify the critical technologies of the blockchain and propose a hash-chain-based cross-regional security authentication scheme for space–air–ground integrated VANETs. We replace the traditional consensus mechanism with Kafka distributed message processing to implement the verification and backup of the records on the block.
3.  Combined with SAGIN to meet the needs of highly intelligent assisted driving, the hash-chain template is redefined according to the characteristics of network transmission.
4.  Using OMNet ++, SUMO, and Veins joint simulation platforms to generate transactions for cross-regional traffic flow, Kafka [11], and Practical Byzantine Fault Tolerance Algorithm (PBFT) [12] consensus algorithms are compared. The experimental results show that the average delay of a single transaction generation block is about 0.9ms, which meets the needs of efficient and low-latency authentication.

The rest of this paper is organized as follows: Section 2 discusses related work. In Section 3, the redefined hash-chain template is introduced. Section 4 describes the model of cross-regional identity authentication system based on the hash chain. Section 5 illustrates the experiment process. Finally, Section 6 concludes this work.

## 2. Related Work

### 2.1. Traditional Cross-Regional Authentication Network

The traditional intelligent transportation system usually has a three-layer or four-layer structure [13]. As shown in Figure 1, the order from high to low is certificate authorities (CA), regional security manager (SM), RSU, and vehicle node. The RSU can also undertake the role of the SM.

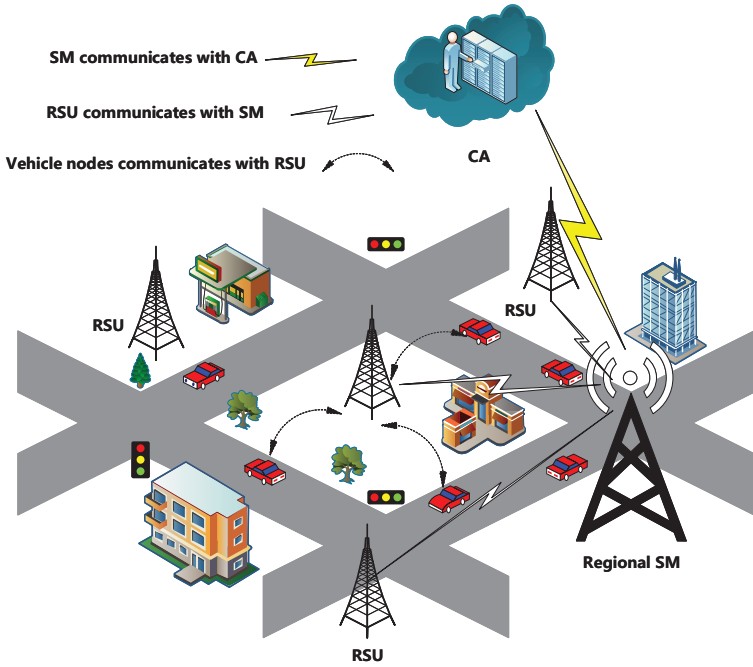

**Figure 1.** Traditional intelligent ground transportation system structure.

Among them, CA has the highest management authority, which is used to issue vehicle identity information and related certificates, verify identity, and manage vehicle pseudonyms. The area managed by the regional SM node is called the security domain and is mainly responsible for information encryption and resource scheduling. A single CA is responsible for multiple SMs, and information exchange between SMs under the same CA needs to be processed by the CA. Located on the third layer is the RSU, which itself acts as a relay node to provide vehicle information with the upper network node when communicating. At the same time, there are also RSU nodes directly under the jurisdiction of the CA as a regional SM node.

However, the intelligent transportation system based on central management has some problems when the vehicle cross-regional identity authentication. On the one hand, due to the limitation of the communication range, the key information of the vehicle needs to be processed by the central management agency during the cross-regional conversion, causing a high delay. Especially in the SAGIN, the vehicle has the characteristics of fast movement, frequent access, and disconnection while driving, which requires continuous identity authentication, so the requirements for delay are more stringent. On the other hand, the CA performs the binding of the vehicle's identity and public key. The attacker can forge the identity and send false information to realize the witch attack. Once the SM node is attacked, it may cause serious security accidents, such as the collapse of the central server.

In response to the problems of the traditional network in the cross-regional identity management of nodes, some scholars have proposed to use blockchain to establish a decentralized network system to avoid loopholes in vehicle certification to ensure the safety and normal operation of the VANETs.

### 2.2. Blockchain-Based Security Authentication

The high-speed mobility of nodes in SAGIN makes the network topology highly dynamic [14]. If further use, centralized key management may cause channel congestion or paralysis during key distribution. Therefore, the researchers consider adopting a distributed network structure to provide corresponding support for the development of SAGIN based on blockchain technology.

With the rapid development of blockchain technology, new exploration based on blockchain technology has attracted extensive attention [15]. Blockchain technology originated from the Internet finance, such as the well-known Bitcoin [16], Ethereum [17], Super Ledger [18], etc. The blockchain can also be used as a decentralized distributed database, with the characteristics of sharing, non-tampering, traceability, and final consistency. Therefore, the application of blockchain in other industries is also developing rapidly. Sun et al. [19] discussed the key technologies and elements that support the development of smart cities based on blockchain shared services. Moreover, the combination of blockchain technology and the Internet of Things (IoT) development has also become a popular application. Ali Dorri et al. [20] summarized the challenges and solutions of blockchain in IoT applications. Based on the Peer-to-peer security and privacy of the blockchain, a new private lightweight blockchain architecture based on IoT security is proposed. To effectively eliminate the high overhead of the blockchain while maintaining its security and privacy, it provides new ideas for the development of IoT applications.

As a special IoT with high-speed mobility, VANETs have a distributed topology structure similar to the blockchain network structure. Therefore, the applications of blockchain in the VANETs have also developed rapidly. Blockchain technology provides solutions based on security, transparency, and data management for the development of VANETs [21]. A lightweight authentication method based on blockchain for the communication security problems faced by intelligent transportation systems is proposed in [22]. The use of distributed vehicle access and revocation schemes instead of the central management digital certificate method can effectively reduce the computing cost and system security response delay. A blockchain-based architecture for Internet smart car ecosystem security and user privacy is proposed in [23]. Wireless remote software updates and other new services (such as dynamic vehicle insurance fees) illustrate the effectiveness of the architecture. We can see that the blockchain technology will have a broader application space in the future, especially in the VANETs.

In the SAGIN, due to the security requirements such as the legitimacy of node identity and data privacy, key blockchain technologies can be combined to improve the reliability of the overall network. However, SAGIN's high-speed mobility and low latency pose new challenges to blockchain technology. Traditional blockchain technology, such as Bitcoin, takes about 10 min to generate blocks, and as the number of block nodes increases, the time and computing power required by the consensus mechanism increase exponentially. This cannot meet the requirements of SAGIN. Therefore, by simplifying the blockchain structure and building a consensus mechanism based on the hash chain, this paper provides support for SAGIN's data transmission and security.

### 2.3. Physical Layer RF Fingerprint Features

Due to security threats such as counterfeiting, eavesdropping, and tampering faced by wireless communication networks, it is crucial to ensure their security performance. However, the security protection measures of wireless networks mostly cover the link layer and above, such as common authentication and authorization or encryption of key information. These security protection measures usually face problems such as complex calculations and key leakage. Therefore, the physical layer security protection has become a new security mechanism [24]. Specifically, physical layer security authentication can use wireless signals to reflect the RF characteristics of the source and analyze the wireless device's communication signals to obtain the device's RF fingerprint for individual identification. Due to the tolerance of the electronic components of the transmitter [25], even the hardware parameters of the wireless devices of the same model and the same batch will be different. This difference is directly reflected in the signal emitted by the device, i.e., the RF fingerprint. It follows

the four principles of universality, uniqueness, collectability, and stability [26]. Universality means that every device capable of wireless communication has corresponding characteristics. Uniqueness means that this feature has nothing to do with the signal modulation method, rate, modulation information, etc., and is determined by the unique physical characteristics of the device. Collectability means that RF fingerprint features can be quantitatively measured. Stability means that the features are not subject to drastic changes such as temperature and environmental changes, and remain stable for a period of time.

Therefore, due to the uniqueness and nonclonability of RF fingerprints, they can become the unique identification of vehicle nodes.

## 3. Hash-Chain

### 3.1. Structure of the Hash Chain

As a simplified structure of the blockchain, the hash chain uses cryptographic encryption to calculate the data blocks linked before and after. Each block is composed of two parts: a block header and a block body. The content of the block header contains information that forms the link relationship between the front and back blocks. The block body is mainly used to store transaction data and transaction details. Combined with the transmission characteristics of SAGIN, we have redefined the hash-chain templates, namely block template and transaction template [27].

The block template is shown in Figure 2a, which contains the local number of the generated block, the identity materials of the source SM ($R_{this}$), the hash value of the previous block, timestamp, and the Merkle root [28]. Among them, collect the wireless signal sent by the SM node and extract the RF fingerprint characteristics of the signal of the device, and use it as the $R_{this}$. The hash value of the previous block is used as the critical data of the block information connected before and after the hash chain. Moreover, the introduced timestamp can be used as an essential basis to ensure the authenticity of the data and traceback at the same time to ensure that the information cannot be tampered with. The hash value, timestamp, root hash, and identity signature of the previous block together generate the hash value of the current block. The Merkle tree structure uses the primary transaction data as the leaf nodes of the hash chain and is constructed in a bottom-up manner. The hash value of the two transactions is calculated to obtain the hash value of the intermediate node. Then the intermediate node hashes in pairs and continues to build up. The final hash result is used as the root hash value of the Merkle tree. Each set of transaction data sets corresponds to a unique Merkle root.

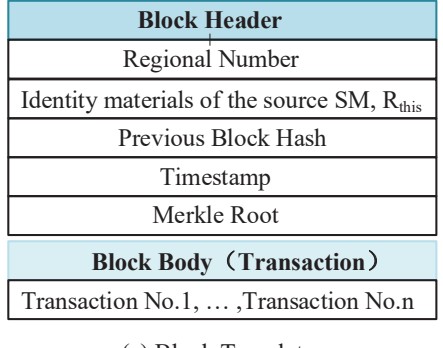

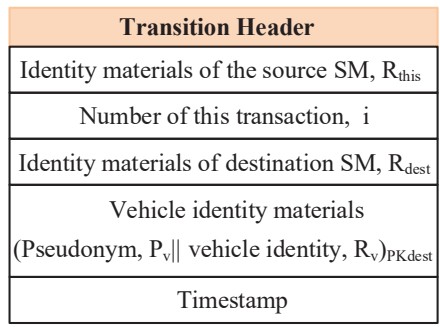

(a) Block Template

(b) Transaction Template

**Figure 2.** Block template and transaction template.

The transaction template is shown in Figure 2b, which is different from the transaction information used for financial management such as Bitcoin and Ethereum. In the SAGIN, the cross-regional identity information exchange of vehicle nodes is used as a transaction content. The identity materials of the SM node in the source area and the destination area is represented by the RF fingerprint. The transaction number is the order of the number of transactions collected in a unit of time. The identity materials

encrypted by the vehicle is to ensure the confidentiality of the information. We use the public key of the SM node in the destination area to encrypt. Other SM nodes cannot read the vehicle identity materials. In contrast, the destination area SM node can use the private key to decrypt to obtain the vehicle identity materials about to enter the area. The SM node packs transactions collected in unit time to generate a block and implements a simplified consensus mechanism to verify and back up the records on the block.

*3.2. Node Function and Role Classification of Hash-Chain System*

In the hash-chain system, nodes have different functions according to the degree of resources scheduled. As the complexity of the system continues to increase, the tasks and permissions of the nodes also have a more evident division of labor. The system functions possessed by hash-chain nodes include bookkeeping function, data storage, query function, transaction submission function, routing function, and member management function. In a traditional blockchain system, the bookkeeping rights of nodes are confirmed through a consensus mechanism, and the common consensus algorithms have proof of work (PoW) [29], Proof of Stake (PoS) [30], PBFT, etc.

Each node in the network is divided into different roles according to the differences in function. The permissions obtained by different characters are also different. In the entire network, the characters responsible for maintaining the operation of the system are mainly the following types:

(1) Full node: The full node keeps a complete and up-to-date copy of the account book, has data storage and query function, a routing function, and an accounting function. It can independently verify all transactions without the help of other node data support. Due to the large amount of ledger data in the entire network, full node requires abundant storage resources and computing resources. Therefore, in the SAGIN, a node with substantial computing power and sufficient storage resources in the ground network can be used as a full node.

(2) Simple payment verification (SPV) node: The SPV node is also called a "lightweight node". Unlike the full node, the SPV node does not have a complete copy of the ledger and only stores block header data, which significantly reduces the overhead of storing resources. During transaction verification, through the "SPV" method, the complete transaction information required from all nodes is synchronized to minimize the time and load of verifying transactions. In the SAGIN, nodes such as drones can be used as SPV nodes due to limited battery and storage resources.

(3) Accounting node: The accounting node is a node that writes blocks in the hash chain through a consensus algorithm, and has accounting function, data storage function, and routing function. The accounting node considers using a simplified consensus algorithm to calculate the hash value of the block in a trusted environment, packaging, and maintaining the block.

(4) Sorting node: The sorting node is to verify and sort the transaction data. By receiving the wireless signal feature identity of the cross-regional vehicle node, the reliability of the vehicle node is confirmed, and the transactions are sorted at the same time to ensure that the hash-chain node records the deal with the same logical order.

## 4. Cross-Regional Identity Authentication Model Based on the Hash Chain

The cross-regional identity authentication model based on the hash-chain proposed in this paper uses the wireless signal RF fingerprint feature as the node identity to ensure the uniqueness and non-cloning. Its structure is shown in Figure 3.

This framework of the SAGIN based on the hash chain is a decentralized distributed structure. The CA is only responsible for the registration of the vehicle nodes when it enters the SAGIN, which need not communicate with the regional SM node. In this paper, a drone with computing power is used as a SM node to generate blocks to ensure the data security of vehicle identity information. In fact, the SM node only needs to have a certain computing and storage capacity. In practical applications, we recommend the use of drones in the background of space–air–ground integrated. The low-altitude flight function of the drones can increase the coverage of the SM nodes, and fewer SM

nodes contribute to the improvement of safety. At the same time, the high degree of flexibility of the drone helps to schedule resources during congestion to help reduce the overall delay of the network. Among them, the SM node of the safe region A adds the cross-regional vehicle identity materials to the blockchain, and the SM node of the safe region B synchronizes the ledger information to obtain the vehicle identity materials entering the area. After the introduction of space-air nodes, the transmission of information through air nodes can reduce the overall delay of the network.

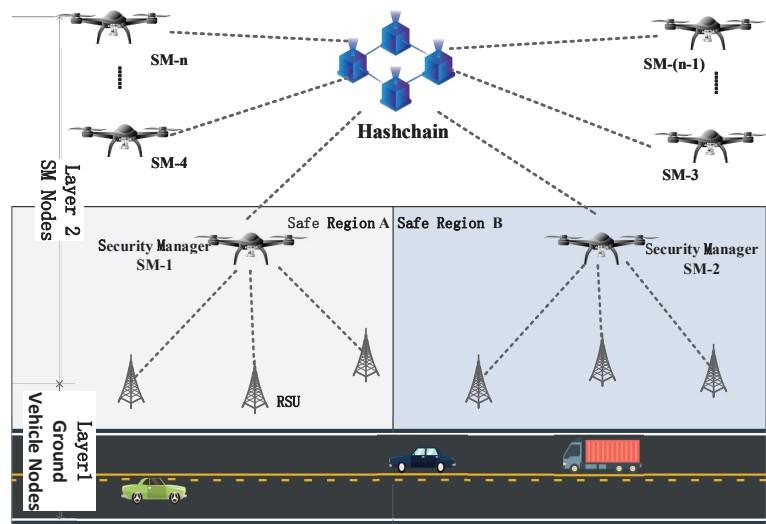

**Figure 3.** Cross-regional identity information management structure based on the hash chain.

*4.1. Authentication Process*

In the case of congested intersections or limited ground communication resources, vehicle nodes can use SAGIN to request more resource support from air SM nodes such as drones and airships. The SM nodes provide resource services for legal vehicles. The essential thing in this process is to ensure the identity security of mobile nodes such as vehicles. This paper compares and analyzes the node identity authentication process of traditional centralized network structure and distributed network structure based on the hash chain, as shown in Figure 4.

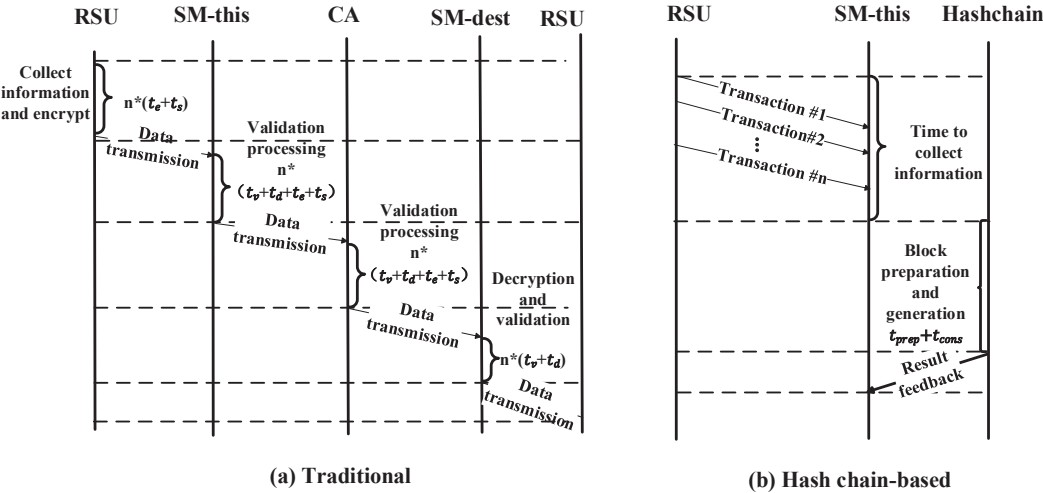

**Figure 4.** Cross-regional authentication process under different network structures.

When a vehicle node crosses from one region to another, that is an area managed by different SMs. The authentication process is as follows.

The cross-regional authentication process of traditional:

Traditional blockchain-based cross-regional activities are packaged according to the number of cross-regional nodes collected per unit time and sent to the corresponding RSU for verification. The corresponding RSU uses SM-this's public key for encryption. When SM-this receives the corresponding data packet, it decrypts and verifies it. If it is legal, the vehicle's identity materials are packaged and transmitted to CA. After CA verification, it is sent to SM-dest. The SM-dest node decrypts and verifies the vehicle identity and transmits it to the corresponding RSU to ensure the security of the vehicle nodes identity that enter across the zone. It can be seen from the above authentication process that a total of three encryption and decryption and signature authentication processes are required. At the same time, it needs to go through four times of waiting and transmission process. Therefore, the overall delay is as follows:

$$T_{tradition} = n * (t_v + t_d + t_e + t_s) * 3 + t_{transmission} * 4 \tag{1}$$

Among them, $n$ is the number of cross-regional requests collected in a unit time. $t_e$ represents the encryption process. $t_d$ represents the decryption process, elliptic curve integrated encryption scheme [31] (ECIES) can be used in this process. $t_v$ represents the verification process. In addition, $t_s$ represents the signature activity, which is mainly used in the elliptic curve digital signature algorithm [32] (ECDSA). $t_{transmission}$ means waiting and propagation delay.

The cross-regional authentication process of hash chain-based:

Before the vehicle nodes enter SAGIN for the first time, the CA completes the registration of the vehicle nodes. At this time, the CA is only responsible for managing the registration and periodic update or distribution of vehicle identities ($R_v$) and does not participate in the subsequent authentication process. During the registration process, we use the USRP to collect the wireless signal and then extract the RF fingerprint features of the vehicle node as its identity. For the identity authentication when the vehicle cross regions, on the one hand, the drone node needs to verify the identity information of the vehicle node, on the other hand, the authenticity of the cross-regional message must also be verified between the drone SM nodes. The specific process is shown in Figure 5, and the process is shown below.

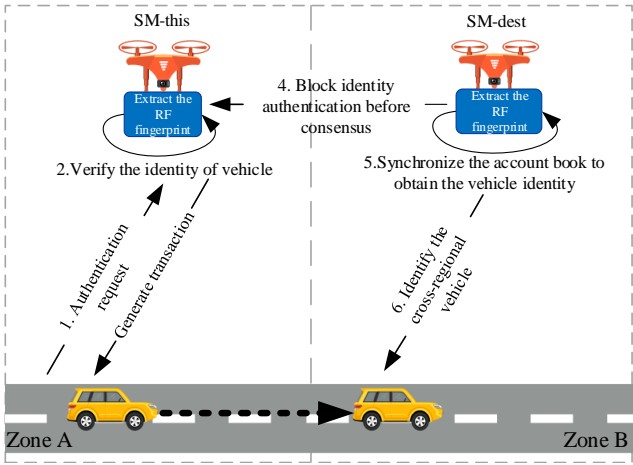

**Figure 5.** The process of cross-regional RF fingerprint identification of vehicles based on the hash chain.

(1) The vehicle node $V$ enters the sky-space integrated network for the first time and sends an identity authentication request to the regional SM node SM-this: $Rv||beacon$.

(2) The SM-this uses the onboard USRP module to collect the wireless signal of the vehicle and extracts its RF fingerprint $R_v{}'$. Compare whether the collected $R_v{}'$ and the $R_v$ sent by the vehicle are consistent. If they match, the vehicle is judged to be a legal vehicle; otherwise, it is an illegal vehicle.

(3) The SM-this determines the vehicle's cross-regional movement by collecting beacons messages sent by legal vehicles, which contain information such as speed, direction, and position. Collect *n* vehicles cross-regional movement per unit time to form *n* transactions. The transaction content includes the identity information of the SM-this, the transaction number, the identity of the SM-dest, the encrypted vehicle materials and the timestamp.

(4) The SM-dest extracts the real-time RF fingerprint of SM-this, and compares it with the RF fingerprint features during registration, so as to judge the legitimacy of the identity of SM-this.

(5) If it is legal, the SM node packages *n* transactions to create a block, and uses the Kafka consensus algorithm to add it to the hash chain and feedback the block result to the SM-this. At the same time, other SM nodes synchronize the contents of account books to obtain relevant information of cross-regional vehicles.

(6) The SM-dest in the destination area obtains the identity of the vehicle nodes that will enter the new area according to the content of the transaction information on the block. If the information exists in the local ledger, it is legal, otherwise it is determined to be illegal and SM-dest refuses to provide services for it.

In the study of individual identification using RF fingerprints, many scholars have conducted a lot of experiments on feature extraction. In this paper, the auxiliary classifier Wasserstein generative adversarial networks proposed in [33] is used to extract the RF fingerprint features of the wireless signal transmitted by the node and identify it. The deep learning-based method avoids the tedious manual design feature process. After the model is trained, the node identity can be quickly identified, and the accuracy rate can reach more than 95%. Compared with the traditional public-private key scheme, the RF fingerprint features of wireless signals is unique and inoperable.

For the hash-chain-based authentication process mentioned above, the time delay includes: first, the time delay of the cross-regional transaction formed by verifying the identity of the vehicle nodes that need to cross the region per unit time. The second is the delay of packaging transactions into blocks and joining the hash chain. The third is the delay of multiple transmission processes. The specific delay formula is as follows:

$$T_{hash-chain} = n * t_{identity} + (t_{prep} + t_{cons}) + t_{transmission} * 2 \tag{2}$$

*n* represents the number of vehicle nodes cross-regional transactions collected in a unit time, $t_{identity}$ represents the time delay for the SM node to verify the vehicle's identity, i.e., the RF fingerprint authentication time delay. $t_{prep}$ and $t_{cons}$ respectively represents the preparation delay of block generation and the delay required to reach consensus. $t_{transmission}$ means waiting and propagation delay.

According to the hash-chain-based authentication process, we conducted a security analysis for the space–air–ground integrated VANETs. Table 1 lists some of the security requirements and corresponding solutions.

**Table 1.** Security requirements and corresponding solutions in the space–air–ground integrated VANETs.

| Requirements | Solutions |
| --- | --- |
| Authenticity | The RF fingerprint of the wireless signal is unique and non-cloneable, ensuring a reliable node identity. |
| Data security | The data recorded on the hash chain have the characteristics of traceability, non-tampering, and non-repudiation. |
| High efficiency | Improve network throughput and reduce block generation delay through Kafka distributed message queues. |
| Against attacks | In the hash-chain system, because the hash value of the block is different from other nodes, it can be discovered in time when the node is attacked; even if the node is broken, there is a backup of the data about the node. |
| Privacy protection | Use asymmetric encryption technology to protect the critical information of the block |

### 4.2. Consensus Algorithm

The traditional Bitcoin system requires about 10 min to generate blocks, which obviously cannot meet the low latency requirements of cross-regional authentication in SAGIN. Therefore, we simplified the consensus algorithm of the traditional blockchain and implemented Kafka distributed message processing instead of PoW, PoS, and other consensus mechanisms to achieve the verification and backup of records on the block. This article compares and analyzes the advantages and disadvantages of the two consensus mechanisms of Kafka and PBFT, and performs simulations in subsequent experiments. Kafka consensus selects several fixed nodes to implement the Kafka cluster to maintain partition logs, and the remaining nodes are used as transaction production and consumers to manage messages in the queue. The advantage of the Kafka consensus is that it has the characteristics of high throughput and low latency. The disadvantage is that it can only tolerate 1/2 of the maximum node failure, and cannot tolerate the existence of malicious nodes. The PBFT algorithm contains a total of three stages, the pre-preparation, preparation, and committing, which realizes the Byzantine problem in the case of a limited number of nodes. The advantage of PBFT consensus is that it can tolerate 1/3 of the failed or malicious nodes, which significantly improves the network's ability to defend against attacks. The disadvantage is that as the number of nodes increases, the broadcast cost also increases, the time complexity of PBFT is as high as $O(n^2)$, and the delay also increases exponentially.

## 5. Experiment

In this section, simulation experiments are conducted based on the distributed network architecture proposed above. OMNet ++, SUMO, and Veins co-simulation platforms are used to generate cross-regional traffic transactions. Through the simulation of network parameters, the cross-regional transaction process of the car flow under the virtual map and the real map is compared. For the simulation of the hash-chain system, based on the Hyperledger–Fabric architecture, we conduct simulation experiments on Kafka and PBFT consensus algorithms to analyze and verify the delay of hash block generation.

### 5.1. Traditional Vehicle Cross-Regional Authentication Simulation

This paper mainly considers the communication and authentication delay between ground vehicles and low-altitude SM nodes. As shown in Figure 6, Veins is an open-source IoT simulation framework, SUMO is used to simulate traffic flow, and OMNeT ++ completes network communication simulation.

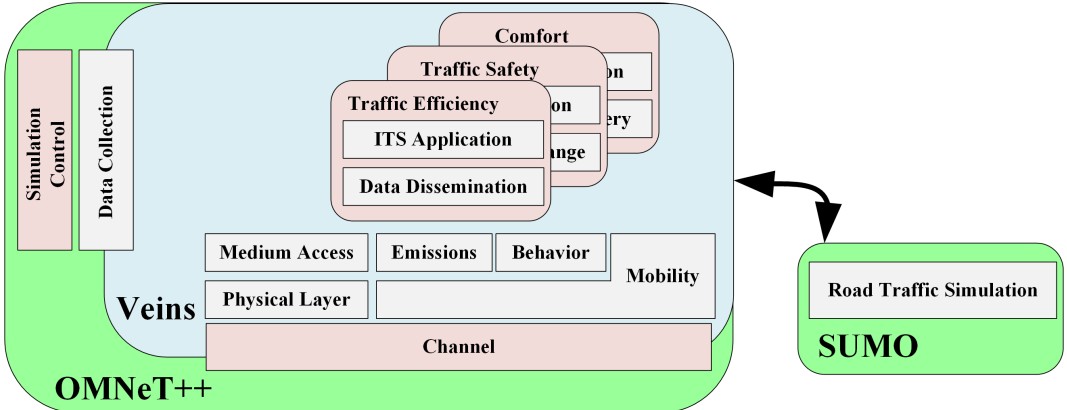

**Figure 6.** Simulation platform diagram.

### 5.1.1. Simulation Parameter Setting

In this simulation, nodes use IEEE 802.11p to communicate, and the data rate is 6 Mbps. The frequency of the car signal is 5.89 GHz. The receiving gain of the car antenna is set to 2.0 dBi, −4.0 dBi, 0.9 dBi, 1.1dBi in the order of front, back, left, and right, and the receiving sensitivity is

−89 dBm. There are two types of simulation scenarios, using a simulated virtual rectangular map and a real map with actual road conditions. The virtual map is shown in Figure 7a, where the RSUs are equidistant, and the communication range of a single RSU is set to 1km, which ensures that the RSUs can interact while covering the entire road. The channel fading model chooses multipath Rayleigh fading. The real map is shown in Figure 7b, adding actual road condition information and obstacle shadow occlusion, and testing network parameters by randomly generated traffic to communicate with the SM node. This experiment introduces the sending source and destination to distinguish the SM node and the vehicle node and adds the vehicle information table to find the illegal vehicle. According to the above settings, we analyze the average value of the results of multiple experiments. Last but not least, in this section, we use the vehicle, cross-regional information, and timestamp to generate record information as the transaction content, so that subsequent hash-chain simulation results can be analyzed for authentication delay.

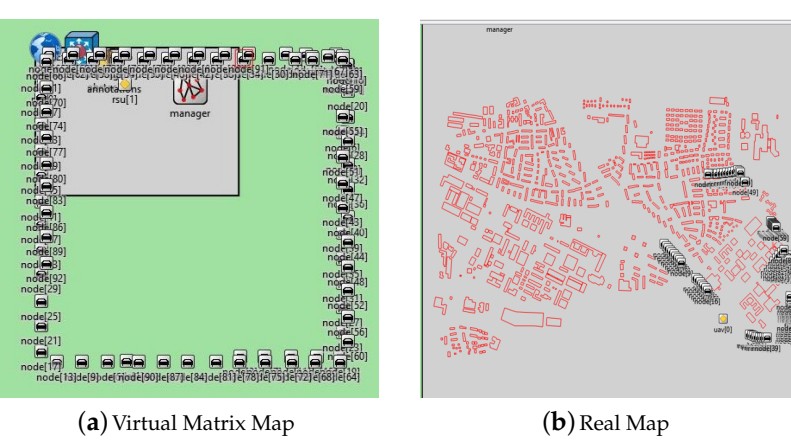

(**a**) Virtual Matrix Map    (**b**) Real Map

**Figure 7.** Simulation scenario.

### 5.1.2. Simulation Results

For the above two experimental scenarios, 20, 40, 60, 80, 100, and 120 vehicles were generated at random positions at the speed of 8.33 m/s (30 km/h), 13.89 m/s (50 km/h), 19.44 m/s (70 km/h) and 27.78 m/s (100 m/s). These cars follow the road and send interactive information with the SM node to update information such as their location in real time.

Compare the transmission delay simulation results of cross-regional identity authentication in virtual map and real map, as shown in Figure 8. It can be seen that the communication delay gradually increases with the increase in the number of vehicles and the vehicle speed, which is positively correlated. Due to the obstruction problem in the actual environment, which prevents communication, the overall delay of the virtual square map is slightly lower than that of the real map, but the difference is not much. When the number of vehicles is between 60 and 80, the increase in delay is not apparent, and even tends to decline. Consider that the network throughput is within the acceptable range when the number of vehicles is between 60 and 80 so that the delay change is not apparent. The transmission delay in Figure 9 is the $t_{transmission}$ in Formulas (1) and (2). The work in this section lays the foundation for the subsequent comparison between traditional authentication and hash-chain-based authentication.

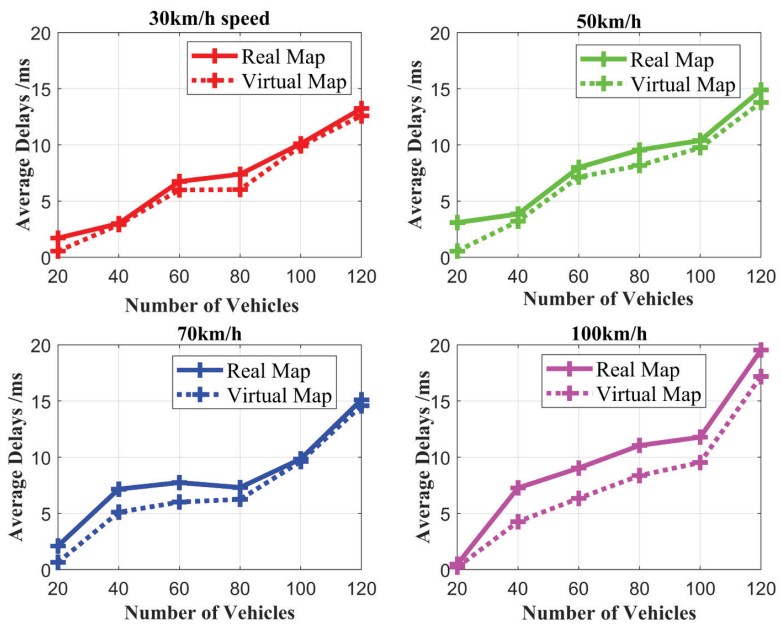

**Figure 8.** Transmission delay of vehicle cross-regional identity authentication.

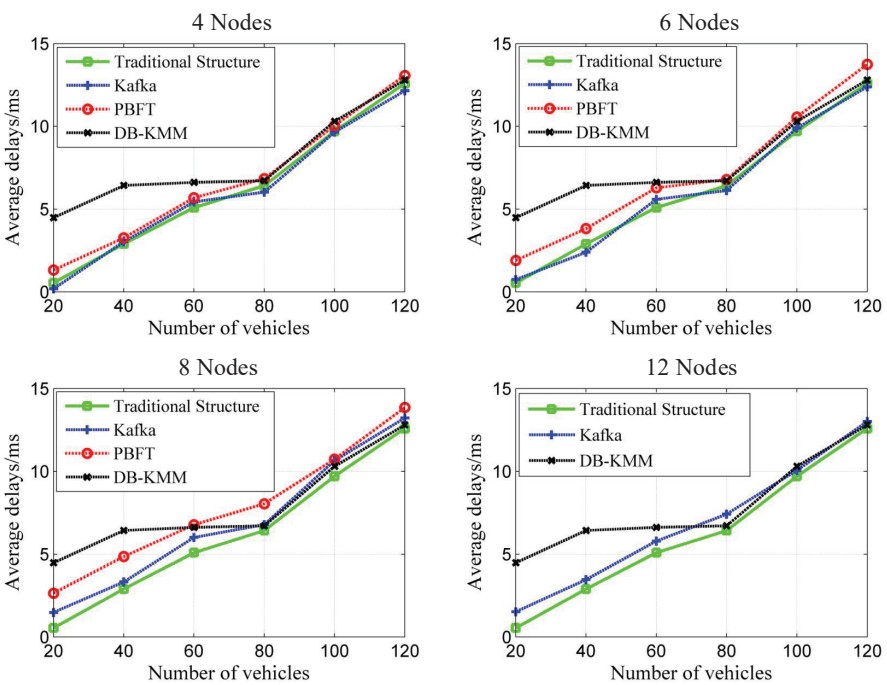

**Figure 9.** Delay of different algorithm.

*5.2. Hash-Chain-Based Vehicle Cross-Regional Authentication Simulation*

In this section, the simulation experiment platform for distributed networks is built on the Hyperledger–Fabric architecture. According to the transaction records generated by the cross-regional traffic flow in the previous section. On the one hand, using the computing network enhanced sn1ne series III, 4-core, 8GB memory Alibaba Cloud server as a simulation device, we compare the delay of generating blocks by Kafka consensus and PBFT consensus. On the other hand, we conducted a comparison experiment between hash block authentication and traditional authentication simulation.

### 5.2.1. Distributed Network Based on the Kafka Consensus

We deployed the Kafka server and pre-fixed three virtual servers as ZooKeeper nodes to achieve the Apache ZooKeeper service, ZooKeeper coordinates each Kafka agent node and other Kafka agents. We use four virtual servers to form a Kafka cluster, and preset two organizations and issue different CA certificates to simulate multiple SMs from various sources in the actual process. Moreover, each organization joins a sorting node to communicate with Kafka nodes to achieve node consensus. Then we increase the number of nodes by two at a time, from 4 to 12. Among them, the maximum request delay for Kafka message packing is 2 s, the maximum number of packed messages is 100, the maximum capacity of blocks is 99 MB, and the maximum size of a message is 512 kb. According to the above settings, we conduct 100 simulation experiments for each different number of nodes and take the average value as the experimental result. Part of the delay data in milliseconds (*ms*) for a specific repeated experiment is shown in Table 2.

**Table 2.** Delay based on the Kafka consensus algorithm.

| Number of Nodes | Number of Transactions | | | | | | | |
|:---:|:---:|:---:|:---:|:---:|:---:|:---:|:---:|:---:|
| | 10 | 20 | 50 | 100 | 150 | 200 | 250 | 300 |
| 4 | 26.356 | 25.342 | 27.708 | **33.708** | 25.53 | **31.881** | 26.511 | **34.902** |
| 6 | 27.138 | 26.241 | 25.908 | **34.331** | 25.866 | **33.241** | 28.186 | **33.98** |
| 8 | 27.202 | 25.313 | 26.464 | **36.392** | 25.957 | **36.395** | 25.267 | **36.191** |
| 10 | 27.317 | 26.541 | 26.457 | **37.045** | 26.757 | **37.425** | 26.847 | **36.754** |
| 12 | 26.15 | 26.952 | 26.796 | **38.281** | 28.26 | **38.434** | 27.461 | **40.098** |

The delay data in the above table is the delay required for every ten transactions. For example, the number of transactions equal to ten means the range from 0 to 10, and the number of transactions 100 means the range from 90 to 100 transactions. After the introduction of the Kafka consensus algorithm, when the number of transactions exceeds the maximum number of packaged messages (100) or the waiting time exceeds the maximum request delay of two seconds, the delay appears once a significant increase, while the other delays are about 26 ms. The time required when the number of transactions is an integer multiple of the maximum number of messages is the time it takes to generate blocks, such as the data in bold in Table 2. The delay of the non-integer maximum number of messages is the time it takes for the docker virtual machine to load after the network parameter restarts, which has nothing to do with the delay of generating blocks. In the actual application process, the network restart delay is not necessarily time. After repeating the experiment enough times, the average delay required to restart the network is about 26.5152 ms. As can be seen from the above table data, as the number of nodes increases, the delay of block generation also increases.

### 5.2.2. Distributed Network Based on the PBFT Consensus

For the PBFT consensus algorithm, when the transaction is initiated, the broadcast message step is gradually performed, and two organizations are still used. Initially set two order nodes and progressively increase, the remaining parameters are consistent with Kafka consensus settings. The experimental delay data in *ms* of the PBFT consensus algorithm is shown in Table 3.

**Table 3.** Delay based on the PBFT consensus algorithm.

| Number of Nodes | Number of Transactions | | | | | | | |
|:---:|:---:|:---:|:---:|:---:|:---:|:---:|:---:|:---:|
| | 10 | 20 | 50 | 100 | 150 | 200 | 250 | 300 |
| 4 | 24.778 | 25.53 | 26.942 | **31.318** | 25.733 | **30.238** | 27.42 | **29.311** |
| 6 | 25.305 | 25.27 | 26.511 | **34.866** | 25.981 | **33.238** | 28.902 | **36.238** |
| 8 | 28.181 | 26.15 | 26.952 | **41.434** | 26.066 | **41.216** | 28.864 | **42.271** |

Due to the limitation of the experimental simulation server device, when the number of nodes exceeds 8, the server memory overflows, and the system crashes. Therefore, we introduce the memory ratio parameter and used cloud server devices with a total memory space of 8 GB to conduct simulation experiments. As the number of nodes increases, the change in memory ratio is shown in Table 4.

**Table 4.** Memory ratio of Kafka and PBFT.

| Consensus | Number of Nodes | | | | |
|---|---|---|---|---|---|
| Algorithm | 4 | 6 | 8 | 10 | 12 |
| Kafka | 23.21% | 24.67% | 25.20% | 25.94% | 28.69 |
| PBFT | 23.20% | 41.70% | 78.50% | - | - |

As can be seen from the data in the table above, for the Kafka consensus algorithm, as the number of nodes increases, the memory ratio changes less, and the load on the network is smaller. However, for the PBFT consensus, as the number of nodes increases, the proportion of memory increases exponentially. When the number of nodes reaches 8, the percentage of memory is limited to 80% due to server configuration restrictions. The stage with the highest proportion is committing, which makes the system unable to simulate more nodes. Furthermore, when the number of nodes is small, such as four nodes, the efficiency of PBFT is higher than that of Kafka. However, as the number of nodes increases, PBFT loses this advantage. After averaging the above two consensus simulation results, the average delay of a single transaction generated block is compared. The result is shown in Figure 10.

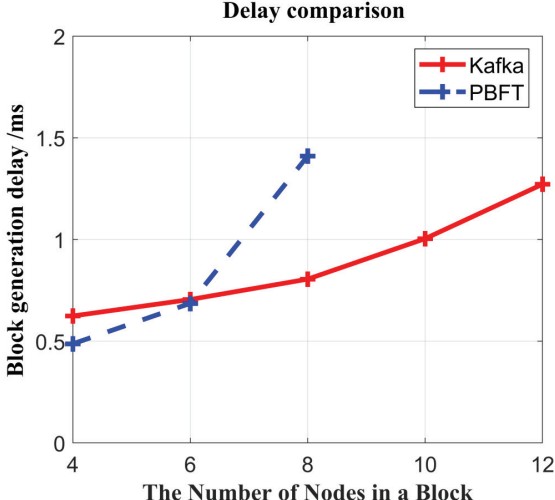

**Figure 10.** Comparison of delays in block generation under two consensus algorithms.

5.2.3. Comparison Experiment of Hash Block Authentication and Traditional Authentication

In the traditional authentication process, we use the ECIES and ECDSA algorithms for encryption and decryption. The average time delay of the two algorithms is shown in Table 5.

**Table 5.** Delay of traditional encryption and decryption algorithms.

| Process | Average Single Processing Delay /(ms) |
|---|---|
| ECIES encryption, $t_e$ | 0.52412 |
| ECIES decryption, $t_d$ | 0.74753 |
| ECDSA signature, $t_s$ | 0.51124 |
| ECDSA authentication, $t_v$ | 1.13411 |

To verify the delay change based on the hash-chain structure and the traditional encryption algorithm, in the case of a virtual map scene with a vehicle speed of 30 km/h, the delay results of the traditional authentication algorithm and the hash-chain authentication algorithm are compared as shown in the Figure 9 below. It can be seen from the simulation results that the delay of the hash chain is increased compared to the traditional authentication algorithm, but the overall difference is smaller, and in some cases, the delay is lower than the traditional authentication algorithm. Moreover, the delay of the hash chain increases as the number of SM nodes increases, which is consistent with the consensus delay verification results. Comparing the two consensus algorithms of Kafka and PBFT, when the number of nodes in the same, the delay of PBFT is larger. In addition, this paper compares the proposed authentication framework with the decentralized key management mechanism for VANET with blockchain (DB-KMM) proposed by Ma et al. [34]. DB-KMM has a higher delay when the number of vehicles is small, mainly from the key management overhead. As the number of vehicles increases, the delay of DB-KMM is similar to the scheme proposed in this paper.

## 6. Conclusions

This article introduces hash-chain technology as part of the new architecture of VANETs. Based on the uniqueness and non-repudiation of physical layer features, the signal fingerprint is used as the identity of the vehicle node. Compared with the traditional central network architecture, it improves network security. The two consensus algorithms were verified and analyzed separately. In the actual process, Kafka's practical efficiency is higher than the PBFT. In the case of a large number of vehicle cross-region transactions that occur in a unit time, the generation delay of the block is allocated to a single cross-region transaction, the average delay of generating blocks is around 0.9 ms. In addition, compared with the traditional encryption authentication scheme, although the overall delay based on the hash-chain structure proposed in this paper has increased, the difference is small, and the security performance has been dramatically improved. Due to the limitation of simulation conditions, the consensus algorithm simulation adopted in this paper has only conducted experiments on some nodes. In future work, it is necessary to measure the quantitative relationship between the number of nodes and the authentication delay, and to study the balance between the number of nodes and the authentication delay in order to improve the authentication efficiency of the entire network.

**Author Contributions:** G.L. built the simulation model and performed the simulation; M.S. wrote the first draft of the manuscript; C.Z. presented the idea and designed the proposed authentication scheme; Z.S. supervised and validated this paper. All authors have read and agreed to the published version of the manuscript.

**Acknowledgments:** The authors would like to express their acknowledgement for the support from the National Natural Science Foundation of China under Grant No. 61971368, the Natural Science Foundation of Fujian Province of China No. 2019J01003, and the National Natural Science Foundation of China under Grant No. 91638204.

**Conflicts of Interest:** The authors declare no conflict of interest.

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
