# Peer review of "Hash-Chain-Based Cross-Regional Safety Authentication for Space-Air-Ground Integrated VANETs†"

_applsci, doi:10.3390/app10124206_

Round 1
Reviewer 1 Report
The work is good and the paper is well prepared. However, there should be shown some comparison with others' achievements.
The authors have compared their algorithms with the traditional and classical ones, and this is fine. However, the algorithms and the experiment results should be seen based on others' techniques and results to show how the authors' methodology outperforms other researchers' methods.
Reviewer 2 Report
The authors study the authentication algorithm for VANETS and propose to use the existing algorithm "Kafka" in their proposed blockchain consensus mechanism.
Their main contribution is the consensus mechanism which can either use "Kafka" or "Practical Byzantine Fault Tolerance Algorithm (PBFT)".
1) The title is misleading since the authors used the term "Space-Air-Ground Integrated". The research domain is VANETs with drones.
2) The RF fingerprint features are mentioned but no analysis or proposed mechanism is found in the manuscript.
3) The authors validate their work by running simulations. Their simulations mostly include ground vehicles and does not cover drones.
4) Simulation results in Figure 7 do not present the performance of the proposed mechanism.
5) In figure 8, the benchmark "PBFT" is shown for only 3 data points and other points are missing? The conclusions made based on figure 8 cannot be validated.
6) In Figure 9, the x-axis label "vehicle number" is misleading. Use the "number of vehicles" as in Figure 7.
7) There are typographical and grammatical errors. "Section 3 title" and "Section 4.2 title". The authors should do a comprehensive English check.
Round 2
Reviewer 2 Report
The authors have addressed my previously raised comments.
Author Response
We are truly grateful to yours critical comments and thoughtful suggestions on our manuscript “Hash Chain-Based Cross-Regional Safety Authentication for Space-Air-Ground Integrated VANETs”,which have significantly helped to improve our work.